# Peritumoral ADC Values Correlate with the MGMT Methylation Status in Patients with Glioblastoma

**DOI:** 10.3390/cancers15051384

**Published:** 2023-02-22

**Authors:** Valentin Karl Ladenhauf, Malik Galijasevic, Johannes Kerschbaumer, Christian Franz Freyschlag, Martha Nowosielski, Anna Maria Birkl-Toeglhofer, Johannes Haybaeck, Elke Ruth Gizewski, Stephanie Mangesius, Astrid Ellen Grams

**Affiliations:** 1Department of Neuroradiology, Medical University of Innsbruck, 6020 Innsbruck, Austria; 2Neuroimaging Research Core Facility, Medical University of Innsbruck, 6020 Innsbruck, Austria; 3Department of Neurosurgery, Medical University of Innsbruck, 6020 Innsbruck, Austria; 4Department of Neurology, Medical University of Innsbruck, 6020 Innsbruck, Austria; 5Institute of Pathology, Neuropathology and Molecular Pathology, Medical University of Innsbruck, 6020 Innsbruck, Austria; 6Diagnostic & Research Center for Molecular BioMedicine, Institute of Pathology, Medical University of Graz, 8010 Graz, Austria

**Keywords:** apparent diffusion coefficient, glioblastoma, MRI, MGMT

## Abstract

**Simple Summary:**

MGMT-methylated glioblastomas have significantly lower ADC values, as compared to the glioblastomas with no MGMT methylation in peritumoral white matter. There were no differences in enhancing tumor areas. These findings could improve predictions of MGMT status in glioblastomas.

**Abstract:**

Different results have been reported concerning the relationship of the apparent diffusion coefficient (ADC) values and the status of methylation as the promoter gene for the enzyme methylguanine-DNA methyltransferase (MGMT) in patients with glioblastomas (GBs). The aim of this study was to investigate if there were correlations between the ADC values of the enhancing tumor and peritumoral areas of GBs and the MGMT methylation status. In this retrospective study, we included 42 patients with newly diagnosed unilocular GB with one MRI study prior to any treatment and histopathological data. After co-registration of ADC maps with T1-weighted sequences after contrast administration and dynamic susceptibility contrast (DSC) perfusion, we manually selected one region-of-interest (ROI) in the enhancing and perfused tumor and one ROI in the peritumoral white matter. Both ROIs were mirrored in the healthy hemisphere for normalization. In the peritumoral white matter, absolute and normalized ADC values were significantly higher in patients with MGMT-unmethylated tumors, as compared to patients with MGMT-methylated tumors (absolute values *p* = 0.002, normalized *p* = 0.0007). There were no significant differences in the enhancing tumor parts. The ADC values in the peritumoral region correlated with MGMT methylation status, confirmed by normalized ADC values. In contrast to other studies, we could not find a correlation between the ADC values or the normalized ADC values and the MGMT methylation status in the enhancing tumor parts.

## 1. Introduction

Glioblastoma (GB) is the most common and most aggressive primary brain neoplasm. The prognosis of GB is very poor, which is often explained by the molecular heterogeneity of its genome, which leads to an unpredictable clinical course in treatment response [1,2,3,4]. The most well-known gene alterations according to the recent World Health Organization (WHO) Classification of Tumors of Central Nervous System include, among others, the genes for the enzyme methylguanine-DNA methyltransferase (MGMT). Modifications of this gene have implications on treatment response and prognosis [5,6].

Magnetic resonance imaging (MRI) is widely accepted as the modality of choice for the diagnosis and evaluation of treatment response [7]. GB has an infiltrative pattern of growth and can expand into normal-appearing brain tissue, which is further than the conventional radiological margin. Computer tomography (CT) and standard MRI testing have underestimated the actual tumor extent [8,9]. In order to achieve successful surgical resection in a best-case scenario, the greatest possible extent of the tumor must be resected without injuring nearby seemingly unaffected brain tissue [10].

Diffusion weighted imaging (DWI), in combination with other sequences, is used for the assessment of brain tissue function and physiology. As a subset of DWI, apparent diffusion coefficient (ADC) maps/values represent Brownian motion in water molecules at a sub-voxel level [11]. ADC maps and DWI are technically robust and can be obtained without administration of a contrast agent. Since the extracellular volume fraction is linked to water diffusion and is highly involved in tissue cellularity, tissue edema, and tissue necrosis, DWI and ADC maps are helpful in the initial assessment and post-treatment assessment of GBs [12,13,14,15,16]. Furthermore, when evaluated against MRI results, ADC values offer a unique characteristic, in that they are numeric values (as compared to the Hounsfield units in CTs) that are easily measurable and can be monitored between different examinations.

Studies have shown that low ADC values before treatment were correlated with high cellularity and the overall survival of glioma patients [12,17]. Furthermore, mean ADC values were correlated with overall survival [15]. In the difficult diagnostic dilemma of pseudo-progression vs. radiation-induced necrosis, ADC values have also been used for differentiation [16].

The molecular profiles of tumors of the central nervous system (CNS) have increased in value in recent years, which has been reflected in the the most recent classifications of CNS tumors by the WHO, setting a greater emphasis on molecular profiling concerning tumor sub-types [18,19] and management. Previous studies concerning MGMT status and ADC values have shown correlations with differing results [20,21,22,23,24]. Normalization of ADC values is a process that aims to reduce variations by calculating the de facto ratio of tumor ADC values and normal-appearing white matter (NAWM) using the patient as their own control [25,26].The normalization of ADC values has also shown differing results, though, especially concerning progression-free survival [27,28].

The aim of this study was to investigate if there were a correlation between ADC values (absolute and normalized) and the MGMT status in glioblastoma, with a focus on the enhancing tumor and the peritumoral region, identified in dynamic susceptibility (DSC) perfusion and T1-weighted sequences after contrast administration.

## 2. Materials and Methods

This retrospective study was approved by the Ethical Committee of the Medical University of Innsbruck (AN 5100 325/4.19). Informed consent was obtained by all participants.

### 2.1. Patients

The time period of recruitment was June of 2015 to March of 2018. Inclusion criteria were patients over 18 years of age, newly diagnosed GB with MRI scans before any therapy, biopsy, or surgery, and a post-surgical neuropathological diagnosis, including MGMT status. The time of survival after the initial MRI-study was retracted from the local patient archive. Exclusion criteria concerning image quality were based on general quality standards in MR-imaging, especially assessing for motion artifacts since they could affect the co-registration of the ROI.

Only IDH wild-type glioblastomas were included in this retrospective study. Analysis was performed via immunochemistry in all 42 patients with further DNA-sequencing in 10 patients.

Histopathological data were considered insufficient if there was no assessment for MGMT status (which was due to less representative tumor material) or the analysis was inconclusive (which was due to poor DNA-quality).

### 2.2. Image Acquisition and Analysis

The MR images for each patient were acquired with a 3T MR imaging scanner (Siemens Healthcare, Verio, Germany) using identical acquisition protocols and parameters. The protocol included DWI and ADC maps (b-values were between 0 and 1000 s/mm${}^{2}$) in an axial plane, T2 turbo spin echo in an axial plane, susceptibility weighted imaging in an axial plane, T2-FLAIR in a coronal plane, T1-MPRAGE before and after contrast administration with multi-planar reconstructions in the axial, sagittal, and coronal planes and dynamic susceptibility contrast perfusion in an axial plane. The specific parameters for DWI are included in Table 1.

The obtained MR images of the aforementioned protocol were digitally transferred from the picture-archiving and communication system (Infinitt; Infinitt, Seoul, Korea) to the post-processing program Olea (Olea Medical, La Ciotat, France). After co-registration via overlay in Olea between anatomical images (T1-MPRAGE after contrast) and cerebral blood volume (CBV) of DSC-perfusion, specific regions of interest (ROI) were defined as following: ROI 1 was defined in the center of the enhancing tumor with the highest visible area of perfusion, excluding necrotic components. ROI 3 was defined directly next to ROI 1, outside the enhancing margin of the tumor in the adjacent white matter. ROI 5 was defined 2 cm from the enhancing tumor core in normal-appearing white matter, and ROI 7 was defined as far away as possible from the enhancing tumor core, also in normal-appearing white matter. In order to minimize the heterogeneity of data according to different brain regions [29], we further defined ROI 2 (corresponding to ROI 1), ROI 4 (corresponding to ROI 3), ROI 6 (corresponding to ROI 5), and ROI 8 (corresponding to ROI) as the mirrored areas in the contralateral healthy hemisphere (Figure 1) in order to calculate normalized ADC values. All manually drawn ROI were co-registered with the concordant ADC maps. After evaluation of the measured, mean ADC values in our ROI, four ratios (Q1, Q2, Q3, Q4) of ADC values were calculated. Q1 was equal to ROI 1 divided by ROI 2, Q2 was equal to ROI 3 divided by ROI 4, Q3 was equal to ROI 5 divided by ROI 6, and Q4 was equal to ROI 7 divided by ROI 8. The ratios corresponded to normalized ADC values. To minimize age-related change, special care was observed not to place ROIs in white matter affected by microvascular damage. We also ensured special care when assessing peritumoral edema in the contralateral hemisphere in order to only include ROIs in normal-appearing white matter that was not affected by peritumoral edema. This was achieved by also assessing T2 turbo spin echo sequences and DWI sequences. The selection of the ROIs, analysis, and evaluation of the ADC values were conducted by an experienced neuroradiologist.

### 2.3. Neuropathological Assessment

All patients underwent either biopsy or resection following a baseline MRI. The tissue was sent to neuropathology for further diagnostics. The MGMT methylation status was assessed by pyrosequencing, using the Therascreen MGMT Pyro Kit (QIAGEN, Hilden, Germany). Tumors with a mean methylation percentage of more than 8% were considered to be MGMT methylated [30].

### 2.4. Statistical Analysis

Statistical analysis was performed using R (R Core Team v. 3.6.1).

The Shapiro–Wilk test and a one-sample Kolmogorov–Smirnov test were used to assess the normality of the data, which were presented with quantile-comparison plots and histograms. The data were normally distributed. Student’s *t*-test was used for comparison between the groups. Total *p*-values < 0.05 were considered statistically significant. The results were presented using boxplots. For the significant areas, the receiver-operating characteristic (ROC) curves were created, and the area under the curve (AUC) was calculated.

## 3. Results

### 3.1. Descriptive Analysis

A total of 63 patients were initially included in the study. Of those, 21 patients had to be excluded due to poor image quality, or insufficient or inconclusive histopathological data. A total of35 patients received surgical resections with histopathological confirmations after surgery. A single patient received a biopsy prior to surgical resection. Seven patients received biopsies with no surgical resection afterwards. After exclusion, 42 patients remained (11 female, 31 male; mean age 64.0 years +/− 14.2). A total of 19 patients (45.2%) had MGMT-methylated tumors (group 1), while in 23 patients, (54.8%) the tumors were MGMT unmethylated (group 2).

The frequency and location of the GBs are included in Table 2.

### 3.2. MGMT Status

We could show significantly (*p* = 0.002) higher ADC values in the peritumoral white matter (ROI 3) of the patients with a negative MGMT methylation status, as shown in Figure 2. The mean ADC values in the peritumoral white matter of patients with a negative MGMT methylation status were $1.21\times 10^{-3}$ mm${}^{2}$/s; however, in patients with a positive MGMT methylation status, the mean ADC values were $0.99\times 10^{-3}$ mm${}^{2}$/s.

The difference of normalized peritumoral ADC values was even more pronounced (*p* = 0.0007), as shown in Figure 3.

There were no significant differences in the ADC values of the enhancing tumor regions or the subsequently calculated normalized ADC values (Q1 and ROI 1) that concerned MGMT methylation status, nor the normal-appearing white matter 2 cm (ROI 5), nor even further away (ROI 7) from the enhancing tumor parts, nor the subsequently calculated normalized ADC values (Q3 and Q4). The mean ADC values in the enhancing tumor region of patients with a negative MGMT methylation status were $0.76\times 10^{-3}$ mm${}^{2}$/s, and in patients with a positive MGMT methylation status, it was $0.79\times 10^{-3}$ mm${}^{2}$/s.

The ROC curves were generated for the peritumoral white matter regions (T2) and normalized values of the peritumoral white matter regions (Q2) (Figure 4). The AUC for the T2-region was 0.75, and for the Q2-region 0.79.The cut-off value for the T2-region was 1.055 with the sensitivity and specificity of 0.739 and 0.737, respectively, and cut-off value for the Q2 region was 1.305 with the sensitivity of 0.783 and specificity of 0.737.

### 3.3. Survival Analysis

In this study, the group the patients with MGMT-methylated tumors did not show significantly increased survival rates, as compared to the patients with MGMT-unmethylated tumors. This was expected, as this group was part of a larger cohort with similar results, which was already published and discussed in detail in a different study [31].

## 4. Discussion

Our results indicated that there were significant differences in the ADC values in patients with GB in their peritumoral white matter, correlating with their genetic profiles of their MGMT methylation status. In a study by Ellingson et al., a positive MGMT methylation status was linked to more pronounced peritumoral edema, though these findings were only based on visual impressions (T2 and FLAIR sequences) and not validated via absolute ADC values [32]. Since ADC values are not solely influenced by edema but also by tissue necrosis and tissue cellularity, they may be superior in assessing peritumoral white matter, in our estimation.

The center of the T1 enhancing tumor corresponded to the necrosis area, so we did not expect to find any reasonably explainable differences in these necrotic areas. The analysis of the ADC values of the more peripherally placed ROIs (ROI 5–8) in normal-appearing white matter also did not show any significant differences. This finding was expected, as well, since conventional MRI sequences cannot depict the real, microscopic invasion of normal-appearing white matter [9]. However, peritumoral T2-hyperintense white matter in GB corresponded not only to the vasogenic edema found in other encountered brain-mass lesions, but it also contained significant amounts of tumor invasion areas [33,34]. When combined with other predictors, this correlated with the likelihood of tumor recurrence after treatment [35]. This was why we expected to find the most pronounced difference in exactly these areas with significant neoplastic presence and close proximity to the enhancing and perfused tumor area, but without visually appreciable necrosis. Choi et al. [36] showed the prognostic value in assessing the ADC histogram analysis, but they found no correlation between ADC parameters and MGMT status. In their analysis, they assessed the extent of the complete conventional tumor but did not assess the peritumoral ADC values, nor did they assess or include tumoral perfusion characteristics, which was fundamentally different to our study design. In other studies, lower ADC values have been associated with more malignant tumors and tumors with higher cellularity [37]. It has even been suggested by various authors that the radiological reports of low grade gliomas should include the location of the areas with the lowest ADC values. These low ADC value areas, along with other imaging indices and characteristics, could represent the most malignant parts of these tumors [38,39] and should, therefore, be assessed in the surgical strategy concerning resection or biopsy. Because if the tissue sampling was localized in a part of the mass-like lesion with higher ADC values, final pathological diagnosis could under-grade the tumor, thus influencing therapy options. However, various studies have pointed out the influence of MGMT methylation status on the survival of GB patients [40,41]. In this light, our results were somewhat unexpected since higher peritumoral ADC values in MGMT-unmethylated tumors suggested lower cellularity and lower peritumoral neoplastic infiltration than their MGMT-methylated counterparts.In this study, this finding could be explained by more pronounced apoptotic or necrotic activity in peritumoral areas of the MGMT-methylated tumors, as shown in studies with phosphorous MR-spectroscopy [31].

Recent studies [27,28] have shown different results concerning the usefulness of normalizing ADC values. In this study, normalization slightly increased the statistical significance in the peritumoral ADC values (ROI3 and ROI4). The other evaluated areas and ratios (Q1, Q3, Q4) did not show any significant differences in absolute and normalized values. Since only ADC values of ROI 3 (peritumoral white matter) showed differences that were highly statistically significant, this finding was expected. Normalization slightly increased the statistical significance but, ultimately, did not produce any new aspects of the defined ROIs, so the additional value may solely be in confirmation. In further studies concerning ADC values in the cerebrum, data confirmation may be important, and the normalization of ADC values is neither complex nor prone to mistakes, so it be a reasonable addition.

MGMT methylation status is an important prognostic factor since the level of methylation of MGMT corresponds to the therapeutic effects of chemotherapeutic alkylating agents, such as temozolomide [42]. Although many studies have been conducted concerning the correlation of visual parameters (tumor location and laterality, enhancement characteristics such as ring enhancement) and MGMT methylation status, there is still no broadly accepted consensus [43]. Further advancements in radiological diagnostics could lead to a future where molecular profiles can be predicted on the basis of an initial MRI study, which could have vast implications for treatment options prior to any surgical intervention (biopsy or resection) and potentially for the surgical strategy, as well. Patients who are unfit for surgery or biopsy could significantly benefit from such advancements. Presently, there is still much uncertainty concerning the prognostic value of ADC values without clear results. Our results indicated that extending the scientific scope beyond conventionally accepted tumor boundaries could be an area of interest for evaluation, but further research is warranted. Since ADC maps are usually part of a routine cerebral MRI tumor protocol, there are large datasets available to evaluate. Enhanced analysis using radiomics could be very helpful for discovering further insights into the diagnostic value of ADC values in the diagnostic stage of GB. Ideally, these studies should include an evaluation of the connection of peritumoral ADC values and the epidermal growth factor receptor (EGFR), which has also been commonly reported as mutated in GB [44]. A study in which we assess the peritumoral ADC values prior to any treatment and after treatment could also be of great interest, especially concerning possible alterations as a marker of the treatment response connected to the genetic profile.

### Limitations of the Study

The evaluation of ADC values is a process prone to inaccuracy with low inter-rater reliability, as the values vary regarding the positioning of the ROI. Although we established a very strict model for positioning the ROI based on multiple sequences in order to link anatomical parameters with a high spatial resolution (T1-MPRAGE after contrast administration with 1mm slice thickness) to functional parameters (DSC perfusion and ADC Maps), it was not possible to exclude all bias due to the manual drawing. The slice length of 3 mm in ADC maps was rather small, but it was adequate for the purpose of interpreting diagnostic studies. However, the co-registration with T1-MPRAGE sequences after contrast administration and DSC perfusion maps could have led to further inaccuracy since ADC maps could be affected by a partial volume effect. This was a retrospective study with a rather small sample size (42 patients); in order to gain further insights, we recommend that our results be validated by a larger study, ideally in a prospective setting.

## 5. Conclusions

The ADC values in the peritumoral region correlated with the MGMT methylation status, confirmed by normalized ADC values. In contrast to other studies, we could not find a correlation between the ADC values or the normalized ADC values and the MGMT methylation status in the enhancing tumor. However, with non-perfect sensitivity and specificity, these values could be a part of future prediction algorithms in order to estimate the MGMT status of glioblastomas in vivo.

## Figures and Tables

**Figure 1 cancers-15-01384-f001:**
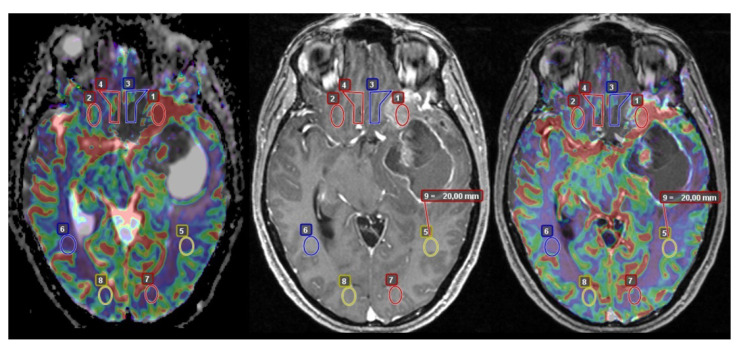
Defined regions of interest in a representative case. Overlay images were acquired in Olea ©. From left to right: Overlay of DSC perfusion and ADC maps; T1 MPRAGE post-contrast administration; and overlay of T1 MPRAGE post-contrast administration and DSC perfusion.

**Figure 2 cancers-15-01384-f002:**
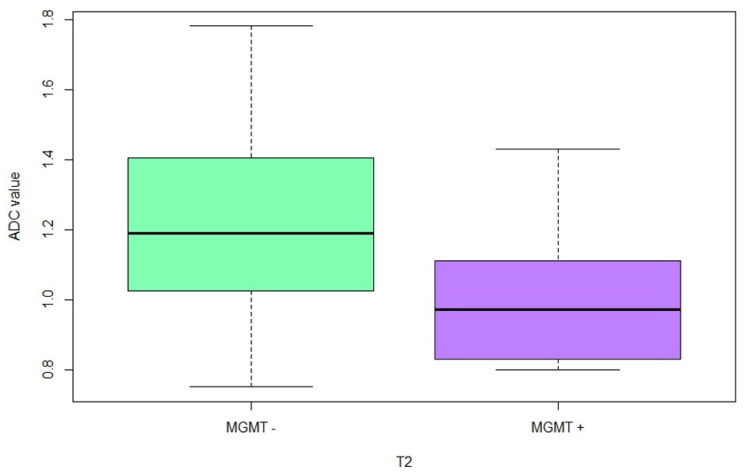
Absolute ADC values in peritumoral white matter (ROI 3) in patients with MGMT-unmethylated GB (MGMT −) and patients with MGMT-methylated GB (MGMT +).

**Figure 3 cancers-15-01384-f003:**
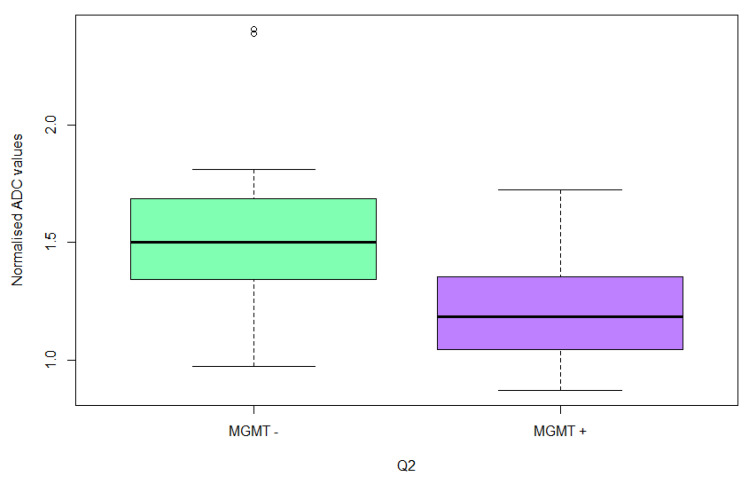
Normalized ADC values in peritumoral white matter (Q2) in patients with MGMT-unmethylated GB (MGMT −) and patients with MGMT-methylated GB (MGMT +).

**Figure 4 cancers-15-01384-f004:**
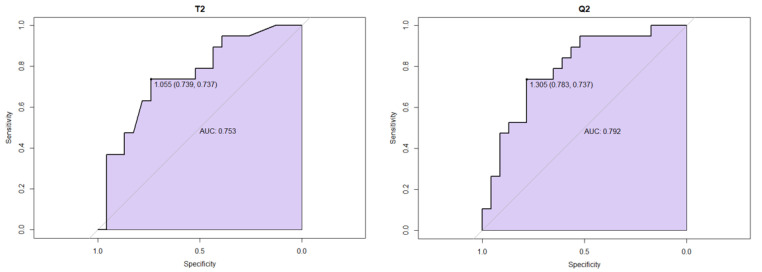
ROC curves for the peritumoral white matter regions (T2) and normalized values of the peritumoral white matter regions (Q2).

**Table 1 cancers-15-01384-t001:** Parameters for ADC Maps.

Parameters for ADC Maps	
Echo time	95 ms
Repetition time	7500 ms
Matrix	256 × 256
Field of view	230 × 230
Time of acquisition	2 min 51 s
Plane	axial
Slice thickness	3 mm

**Table 2 cancers-15-01384-t002:** Location of GBs and frequency.

Location of GBs and Frequency	
Temporal left	11
Temporal right	4
Temporoparietal left	1
Temporoparietal right	1
Frontal left	4
Frontal right	6
Parietal left	5
Parietal right	3
Basal ganglia right	2
Occipital left	3
Temporo-Occipital left	1
Temporo-Occipital right	1

## Data Availability

The original data regarding this study can be made available upon request at the corresponding author.

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
