# Peer review of "Peritumoral ADC Values Correlate with the MGMT Methylation Status in Patients with Glioblastoma"

_cancers, 2023, doi:10.3390/cancers15051384_

Round 1

Reviewer 1 Report

First, I would like to commend the authors for their contribution on this interesting topic in glioblastoma literature. I do have some concerns and some questions.

MAJOR CONCERNS

1/ p 2 of 9; M&M; patients: I strongly suggest that the authors include the time period during which the patients were recruited.

2/ p 3 of 9; M&M; Neuropathologic assessment: The authors rightfully refer to the 2021 WHO classification of CNS tumors. But they do not mention the IDH mutation status of the glioblastoma patients included in the study. I consider this to be crucial information that should be supplied, whether IDH mutation analysis was performed using IHC and/or NGS.

MINOR CONCERNS / QUESTIONS

3/ p 1 of 9; abstract; line 6: The authors state that the MGMT enzyme is methylated. However, it concerns an epigenetic phenomenon: the promoter of the MGMT gene is methylated, hence blocking its transcription, hence resulting in fewer amount of and weaker effect of the MGMT enzyme. 

4/ p 4 of 9; results; descriptive analysis: The number of 69 patients seems rather low. Or did the authors include only the patients of a limited time period (see also question 1); please elaborate. 

5/ p 4 of 9; results; descriptive analysis: 33.3% of the patients initially included had to be excluded. The authors attribute this high number to poor image quality and/or insufficient or inconclusive pathology. Please elaborate on the latter, since this may contribute to bias: e.g. did the excluded patients have MGMTp methylated tumors or was MGMTp status unknown? A short flow chart could be very helpful.

6/ p 6 of 9; discussion; lines 166-170: The finding of lower ADC in the peritumoral zone of MGMTp methylated GBM  is a very interesting finding as is the comment by the authors. Do the authors think that the better prognosis of GBM patients with MGMTp methylated tumors is apparent only if the patients are treated with chemoradiotherapy; in other words, the MGMTp status reflects the potential vulnerability of the tumor to current treatment, but most likely not the aggressiveness an sich of the natural course of the disease (of course, talking only of MGMTp methylation in IDHwt GBM patients)? From this point of view, the finding of the authors in patients before treatment, may be not so surprising. Perhaps this topic merits some further consideration in the discussion. 

Reviewer 2 Report

Dear authors,

I read with interest the study regarding utility of peritumoral ADC values in predicting methylation status. The results show significant difference between peritumoral ADC values.

I have few comments.

Major comments-

1. Can the authors also provide exact mean values of ADC for both enhancing and peritumoral regions.

2. Also, just mentioning significant difference between two may not be enough. Can you pls include ROC AUC analysis to see if there can be a ADC optimal cut off value with sensitivity/specificity.

3. Also, ROI2 was placed using CE images and not ADC or FLAIR images. I am concerned that using these images may miss peritumoral edema. Did authors take any support of FLAIR/DWI images for segmentation of peritumoral edema.

Minor comments

There is some confusion regarding lower ADC values in methylated and unmethylated tumors. Please see your simple summary and lines14-17 in abstract.

You may include few references that did not show any ADC difference between enhancing tumors for methylation status (for example PMID: 23275590)

Author Response

Cover letter for the referee reports for the scientific article:

“Peritumoral ADC–values correlate with the MGMT-methylation

status in patients with glioblastoma”:

Thank you very much for considering our scientific article. We have read the comments/concerns with great interest and believe that we have covered all of them in this file.

As an attachment you will find the revised version of our article, adjusted sections are highlighted in blue.

Please contact us at any time, if there should be any comments or questions concerning our adjustments.

- Dear authors,

- I read with interest the study regarding utility of peritumoral ADC values in predicting methylation status. The results show significant difference between peritumoral ADC values.

- I have few comments.

- Major comments-

- 1. Can the authors also provide exact mean values of ADC for both enhancing and peritumoral regions.

Answer:

In ROI 1 in MGMT not methylated GB the mean values are 0.76 10-3 mm2/s , in ROI 3 1.21 10-3 mm2/s.

In ROI 1 in MGMT methylated GB the mean values are 0.79 10-3 mm2/s, in ROI 3 0.99 10-3 mm2/s.

- 2. Also, just mentioning significant difference between two may not be enough. Can you pls include ROC AUC analysis to see if there can be a ADC optimal cut off value with sensitivity/specificity.

Answer:

Thank you very much for this important suggestion. We have now included ROC / AUC – Analysis in our article in section 3.2.

The ROC-curves were generated for the peritumoral white matter regions and normalized values of the peritumoral white matter regions. The AUC for the T2-region was 0.75, and for the Q2-region 0.79.The cut-off value for the T2-region was 1.055 with the sensitivity and specificity of 0.739 and 0.737, respectively, and cut-off value for the Q2 region was 1.305 with the sensitivity of 0.783 and specificity of 0.737.

- 3. Also, ROI2 was placed using CE images and not ADC or FLAIR images. I am concerned that using these images may miss peritumoral edema. Did authors take any support of FLAIR/DWI images for segmentation of peritumoral edema.

Answer:

Thank you for this important comment. The fact that tumoral edema is influencing ADC – values in the peritumoral region (ROI 2) is part of the discussion (line 135 – 140 and furthermore 146 - 152). In order to place ROI in normal appearing white matter in the contralateral hemisphere (ROI 4), T2 turbo spin echo sequences and also DWI – Sequences were assessed but ROI – placement was ultimately done in contrast enhanced images as described in 2.2. We have now included this information in section 2.2.

- Minor comments

- There is some confusion regarding lower ADC values in methylated and unmethylated tumors. Please see your simple summary and lines14-17 in abstract.

Answer:

This is due to a typing error and was corrected, thank you very much for spotting this mistake.

- You may include few references that did not show any ADC difference between enhancing tumors for methylation status (for example PMID: 23275590).

Answer:

We have now included your stated reference, thank you for expanding our manuscript.

We have also included a second new reference (PMID: 21833736) which did not show significant differences in absolute ADC – values in the enhancing tumors. PMID: 27120357 (Line 185 – 187 in our discussion) also includes a study which shows no differences in ADC – Parameters according to MGMT – Status.

Reviewer 3 Report

In this paper the authors provide new evidence regarding ADC-Values in the peritumoral region and their correlation with MGMT- methylation status in patients with glioblastoma. The article is well-structured, the experimental model and the protocol employes are clearly explained; the authors have been able to concisely and clearly convey the usefullness of their result in discussion section pof the article.

Minor concerns

- The authors reported that 21 patients had to be excluded owing to poor image quality or insufficient or inconclusive histopatological data. It could be usefull to know clearly if there are some exclusion criteria and, in particular, when an histopatological data is considered doubtful.

- The population mean age is 64.0 years +/- 14.2. It could be interesting to know the age of all patients in order to describe better the population itself. It is known that ageing changes the brain, especially as regards microvascular damage. Does the neuroradiologist consider this aspect in the process of ROI- choosing?

All in all, It would be useful to know how many patients underwent biopsy or microsurgery resection and the brain site where patients had the lesion.

This information may be important to better understand the features of the lesions in exam, in particular for future clinical practical applications, considering that, depending on MGMT-methylation, some chemioterapy drugs can have a role also in patients that can’t undergo surgery.

Round 2

Reviewer 1 Report

I wish to thank the authors for the answers to the questions they provided and the adjustments to the manuscript. I believe the publication is a worthy contribution to the glioblastoma literature.